# Surface-Functionalized Separator for Stable and Reliable Lithium Metal Batteries: A Review

**DOI:** 10.3390/nano11092275

**Published:** 2021-09-01

**Authors:** Patrick Joohyun Kim

**Affiliations:** Department of Applied Chemistry, Kyungpook National University, Daegu 41566, Korea; pjkim@knu.ac.kr; Tel.: +82-53-950-5582

**Keywords:** Li metal anode, functional separator, battery safety, separator-coating materials

## Abstract

Metallic Li has caught the attention of researchers studying future anodes for next-generation batteries, owing to its attractive properties: high theoretical capacity, highly negative standard potential, and very low density. However, inevitable issues, such as inhomogeneous Li deposition/dissolution and poor Coulombic efficiency, hinder the pragmatic use of Li anodes for commercial rechargeable batteries. As one of viable strategies, the surface functionalization of polymer separators has recently drawn significant attention from industries and academics to tackle the inherent issues of metallic Li anodes. In this article, separator-coating materials are classified into five or six categories to give a general guideline for fabricating functional separators compatible with post-lithium-ion batteries. The overall research trends and outlook for surface-functionalized separators are reviewed.

## 1. Introduction

Since the advent of commercial rechargeable batteries using the combination of graphite and LiCoO_2_, secondary lithium-ion batteries (LIBs) as practical energy reservoirs have received continuous attention from industries and academics, owing to their decent energy density, long life span, and low self-discharging rate [1,2,3,4,5,6,7]. As state-of-the-art electronic devices such as electric vehicles (EVs) and portable devices become a big part of our lives, the demand for developing more advanced LIBs with high capacity, fast charging ability, and improved reliability has been accelerated to keep pace with this fast-moving trend [8]. Given the specification of the above-described advanced electronic devices and the mission to reduce greenhouse gas emission to net-zero by 2050, it is inevitable to move on from the LIB technology to more advanced battery systems such as Li metal batteries (LMBs), which can dramatically increase the energy density of the existing energy storage technology [9,10,11]. Before the commercialization of LIBs, Dr. Whittingham first developed a battery prototype using a Li metal paired with a TiS_2_ electrode in the 1970s [12]. The developed Li|TiS_2_ cell has an electrochemical potential of about 2.5 V, which is not high enough for achieving high specific energy density. In 1989, Moli Energy commercialized Li|MoS_2_ cylindrical-type rechargeable batteries that were successful in the commercial market for a while [13]. However, owing to the frequent occurrence of safety issues such as catching fire, LMB products eventually disappeared from the market at the time. 

Recently, Li metal anodes have come into the spotlight again due to the limited energy density of existing LIB technology that employ a Li-containing cathode and a carbon-based anode [10,11]. Metallic Li is considered the “holy-grail” anode, owing to its tenfold higher specific capacity than “graphite”, its low material density (0.534 g cm^−3^), and the most negative electrode potential (−3.04 V versus S.H.E. at 25 °C) [10,11]. Furthermore, the adoption of metallic Li as an anode is indispensable to realize high-energy-density batteries such as Li-O_2_ and Li-S cells, both of which are being considered the “future of energy storage” [14,15,16,17,18,19,20,21,22,23,24,25]. Due to these promising prospects, a number of research projects focusing on the Li anode have been extensively carried out. Until now, only a few Li anode systems have shown promising results in terms of electrochemical stability and performance. For example, all-solid-state Li metal batteries with a thin Ag-C layer were successfully demonstrated by Samsung’s research group [26]. The designed pouch cell ran over 1000 cycles with maintaining a high energy density (>900 Wh l^−1^) and an excellent Coulombic efficiency (99.8%). This approach presents a bright outlook where Li-metal systems can be practically adopted as next-generation energy storage. However, there are at least four representative challenges to overcome in order to make all solid-state LMBs commercially available in the market. (1) It is technically unable to form a perfect contact between the solid-state electrolyte and the electrode when fabricating solid-state batteries, which significantly increases the impedance at the electrode/electrolyte interface. (2) Not all the solid electrolytes have a good compatibility with electrodes, thus requiring an additional process to alter the surface nature of solid electrolytes. (3) Relatively slow kinetics of Li-ion transport through the solid-state electrolyte (in comparison with a liquid electrolyte) inhibits the practical use of Li metal anodes in the applications (e.g., EV) requiring fast charging and high power density [27,28]. (4) The manufacturability and material cost of solid-state batteries are not cost-efficient so far, in comparison with that of an liquid electrolyte system [29,30]. In this regard, it would be reasonable to take into account other systems using organic liquid electrolytes and understand what would happen if Li metal is employed in liquid electrolyte-based LIBs. 

Metallic Li with “host-less” nature can be infinitely changing during Li plating/stripping process. This leads to a breakdown/restoration of an unstable and non-homogenous solid electrolyte interphase (SEI) layer over the surface of Li metal electrode [11,31,32,33,34]. During Li plating (or termed ‘deposition’), Li^+^ flux is significantly intensified through the cracks of the SEI layer, giving rise to the formation of sharp Li dendrites [31,35,36,37]. The physical features of the formed Li dendrites could lead to the puncture of a polyolefin separator, which dramatically increases the risk of a battery short-circuit. Apart from the formation of Li dendrites, electrochemically dead and isolated Li pillars detached from the Li metal anode are floating inside the cell, which reduces the actual mass of electrochemically active Li (Figure 1a) [11]. All of these unwanted phenomena become even more serious when LMBs are operating under abuse conditions, i.e., high/low temperature, high current density, and overcharging. Especially, these topics are of importance since they can directly affect the safety of our lives [38,39,40,41]. 

A number of strategies have been proposed to make LMB technology feasible and safer, such as stabilizing the Li electrode surface via interface modification [42,43,44,45,46,47], modifying the chemical constituents of electrolytes [48,49,50,51,52,53,54,55,56,57,58], adding a flame-retardant-based separator [59,60,61,62], and functionalizing the surface of polymer separators [23,63,64,65,66,67]. Encapsulating the Li metal with a protective layer through vapor deposition methods (e.g., physical vapor deposition) has shown a stabilized surface reactivity against atmospheric environments (air and humidity) and has presented optimistic electrochemical results in terms of cycle performance and Coulombic efficiency [43,44,45,46]. Nevertheless, the approach to modify the surface of Li metal via sophisticated techniques is technically hard to achieve both reproducibility and scalability. As to the electrolyte modification, it has somewhat shown impressive effects on suppressing the growth of Li dendrites, enhancing the Coulombic efficiency, and establishing the robust SEI layer [49,50,51]. Moreover, this approach is highly compatible with the conventional battery manufacturing process. Due to these benefits, it is considered the most feasible strategy to resolve the intrinsic problems of Li metal anodes. However, it itself cannot be a fundamental solution to prevent the battery short-circuit caused by Li dendritic growth. In addition, unavoidable side reactions including a parasitic reaction between a Li anode and liquid electrolyte seriously deteriorate the Coulombic efficiency and capacity retention of LMBs [68]. 

Of these approaches, the surface functionalization of polymer separators has been relatively less studied than the other strategies. In addition, the way of modifying the surface of conventional separators with functional materials is an efficient and effective approach to directly regulate Li^+^ flux, control the geometry of Li deposition over the electrode, and alleviate the side reactions of chemical crossover [63,64,67,69,70,71,72]. Figure 1b shows the schematic of a conventional LIB configuration and a LMB configuration introduced with a surface-functionalized separator. The coating layer on the separator can face either the anode side or cathode side according to the specific objectives and requirements of the targeted cell. The recent trends of separator research and the strategies to mitigate the formation of a Li dendrite using surface-functionalized separators are reviewed on the following pages.

## 2. Conventional Polymer Separators for LIBs

Conventional rechargeable batteries are composed of four components: (a) two electrodes, i.e., cathode and anode, (b) a separator, and (c) liquid electrolyte. The battery components of two electrodes and an electrolyte directly participate in electrochemical reactions. In contrast, a battery separator should be electrochemically inactive and mechanically stable while running electrochemical cells [73,74]. The existing roles of separators mainly focus on the basic functions: (a) a physical barrier between anode and cathode, (b) an electrolyte reservoir, and (c) a path for ion transfer. These roles are of crucial importance to facilitate the stable operation of LIBs. In addition to these essential functions, battery separators are required to shift from a passive role to a more active role, especially in advanced LIBs in order to keep up with the research trends with regard to battery safety.

### 2.1. General Characteristics of Separator for LIB and Requirements of Separator for LMBs

#### 2.1.1. Porosity, Pore Size, and Thickness

The porosity, pore size, and thickness are crucial features determining the properties of separators and the overall performances of cells. In general, the pore size and porosity of typical separators is around 1 µm and 40%, respectively [73]. In order to use the separator for practical LMBs, the pore size is required to be small enough (e.g., sub-micrometer) to physically block the penetration of growing Li dendrites, and the porosity should be comparable to typical separators in order to facilitate efficient Li^+^ transfer across the separator. In case of separator thickness, commercially available Celgard’s polymer separators have a thickness of around 25 µm [73]. If the total thickness of the separator gets thicker, it increases the overall resistance of the cell and reduces the loading amount of active materials in the cell. On the contrary, if the total thickness of the separator becomes thinner, it increases the risk of separator puncture by growing Li dendrites.

#### 2.1.2. Electrolyte Wettability

The battery separator should get wet as soon as it contacts liquid electrolytes. It is reported that the electrolyte wettability of conventional separators is highly associated with battery cycle retention and capacity [56,73,74]. If the separator has a poor electrolyte wettability, it would cause a non-uniform Li-ion transport across the separator. This would result in uneven Li deposition over the electrode, consequently leading to a short circuit of LMBs [75]. To facilitate the homogeneous Li deposition/dissolution in LMBs, it is of paramount importance to improve the electrolyte wettability of separators. The wetting behavior between a separator and a liquid electrolyte is typically studied by using the contact angle measurement [76]. If the separator has a good affinity with liquid electrolytes, the angle between the separator surface and the curvature of an electrolyte droplet would be small. On the other hand, if the separator has a bad affinity with liquid electrolytes, it would have a large contact angle. With these measurements, the electrolyte wetting behavior of separators can be directly determined.

#### 2.1.3. Thermal Properties

One of the critical causes of battery thermal runway is thermal abuse [77]. It mostly happens when the contact between the current collector and the electrode is loose. The poor interface connection results in the huge loss of energy, i.e., excessive heat flow, which accelerates the temperature rise of the cell. Due to the relatively poor thermal stability of polyolefin at a high temperature, the polymer separator would eventually experience serious shrinkage, thereby leading to the battery short-circuit [78]. In this regard, the polymer separator should maintain its structure integrity even at elevated temperatures, in order to circumvent the potential battery failure. According to USABC requirements for conventional battery separators, the targeted shrinkage of the separator is less than 5% at 200 °C, and the calendar life of the separator is 15 years [79]. Therefore, the thermal properties of separators should be taken into account when designing functional separators for LMBs.

#### 2.1.4. Mechanical/Electrochemical Stability

For battery assembly, separators need to meet the following requirements: (1) high mechanical strength to endure the tension and pressure during the battery assembly, (2) excellent insulating property in order not to pass electrons across the separator, and (3) decent electrochemical/chemical stability to use the separator continuously for more than 1000 cycles without degradation [73,80,81]. Especially in the LMB system, the separator should be mechanically strong enough in the liquid cell to suppress the piercing of Li dendrites and the expansion of high-capacity electrodes such as Si and Li and to protect the entire cell from the external stress caused by physical shock and pressure. To evaluate these mechanical properties of separators, mechanical abuse tests (such as nail penetration, puncture, flat crush, and edge crush) and interface-bonding analyses are widely implemented across the industries [82,83,84,85,86,87]. 

### 2.2. Technical and Inherent Issues to Be Resolved

Polypropylene (PP) and polyethylene (PE) are two of the representative polyolefins typically used for fabricating conventional separators [74,80]. The separator fabricated by either wet process or dry process has a number of pores throughout the entire area (Figure 2a,b) [73]. Each has good mechanical strength and chemical stability against polar solvents. However, these separators are vulnerable to high temperature and have a relatively poor electrolyte wettability highly associated with the overall resistance in the cell, limiting the practical use for applications (such as EVs) requiring high current levels [73,81]. One of the other serious challenges when a Li metal anode is employed in rechargeable batteries is the separator puncture caused by the penetration of growing Li dendrites. This phenomenon was visually revealed by the transparent cell tests shown in Figure 2c [67,88,89]. Note that extremely high current flows through the path where Li dendrites penetrate the separator, thus resulting in an exothermic reaction inside the cell followed by an explosion (or catching fire) of LMBs (Figure 2d,e) [90]. Typically, the average pore size (e.g., around 1 µm) of polyolefin separators is too large to prevent Li dendrites from piercing the separator. It is reported that the separator with a pore size less than 5 nm has a substantial effect on resisting Li dendrite penetration and redistributing Li^+^ uniformly across the separator [91,92,93]. In order to meet the technical requirements described above and address the potential issues of LMBs, it is of importance to find the optimal condition for fabricating surface-functionalized separators that could maximize the stability and performance of LMBs.

## 3. Approaches to Modify the Surface of Conventional Polymer Separators for LMBs

Among the important components of batteries, the battery separator has received relatively less attention than other parts in order to solve the systemic challenges of the Li anode system. The topic of modifying the surface of separators with functional materials has shown promising results in terms of cycle stability and Coulombic efficiency [63,67]. In addition, it is an efficient and straightforward strategy to control the morphology and growing orientation of Li dendrite. Typically, the coating materials to functionalize the surface of polymer separators can be classified into five or six types: (a) thermally conductive materials [63,64,69,94,95,96,97,98], (b) metals [99,100,101,102], (c) polymers [103,104,105,106,107,108], (d) carbons [66,109,110,111], (e) metal oxides [67,88,112,113,114,115,116,117,118,119], and (f) others [120,121,122,123,124] (Figure 3). For electrochemical evaluation, each separator coating material is laminated onto one side or both sides of the polymer separator, using the tape-casting method, physical vapor deposition, etc. The information of separator-coating materials, separators, fabrication techniques, and electrochemical results is listed in the Table 1.

### 3.1. Surface Functionalization of Polymer Separator with Thermally Conductive Materials

Several nitride materials such as boron nitride (BN) and aluminum nitride (AlN) are known to have high thermal conductivity and decent electrochemical/chemical stability against Li metal in comparison with other metal compounds [63]. Hu et al. demonstrated that a BN layer is able to homogeneously spread the localized heat generated during exothermic electrochemical reaction, contributing to a uniform Li plating/stripping (Figure 4b) [69]. Due to the BN-coating effect, the diameter of formed Li pillars was notably increased. This significantly reduces the risk of a short-circuit of LMBs. Additional experiments such as IR temperature mapping directly proved that the BN layer is able to efficiently reduce the temperature spikes that randomly happen over the surface of the separator, owing to the effective heat-spreading effect. As a follow-up work, Paik et al. deposited another carbon layer to the opposite side of the BN-coated separator and evaluated the designed separator for Li-S and Li metal batteries (Figure 4d) [64]. The similar effect of enlarging the diameter of Li dendrites was observed when the BN layer faced the Li metal anode. In addition, two shielding layers of the fabricated separator efficiently confined polysulfide species within the cathodic area during an electrochemical test. Rodriguez et al. employed a mixture of BN and graphene in order to improve both the thermal conductivity and electrical conductivity of the coating layer (Figure 4c) [97]. It exhibited the enhanced cycle stability and rate capability of LMBs due to the synergistic effects of the composite layer. Pol et al. first attempted to use AlN nanoparticles as a coating material for fabricating a functional separator for LMBs (Figure 4e) [63]. The AlN layer was positioned on the anodic side to see how the AlN layer affects the morphology of the Li metal surface after multiple Li plating/stripping processes. As observed in the works using a BN layer, the AlN layer revealed an analogous effect on increasing the diameter of Li dendrites. In addition, the Li|Cu cell with an AlN separator exhibited an improved Coulombic efficiency by comparison with the control cell. Apart from simply forming either a BN layer or an AlN layer over the polyolefin separator, these materials can be further extended to other types of separators. For example, Hu et al. fabricated a BN-PVDF-HFP separator using a 3D extrusion printing technique [95]. The Li symmetrical cells with the BN-PVDF-HFP separator presented promising results in terms of cycle stability and Coulombic efficiency. Moreover, the dendritic Li growth was effectively suppressed due to the excellent thermal management performance of the fabricated separator. Extrapolating from all these previous studies, a few nitrides are considered to be promising coating materials for controlling the physical features of dendritic Li and managing the thermal behavior of the Li anode surface (Figure 4a). As presented in Figure 4f–h, dendritic Li has dissimilar morphology and diameter when using different types of separators. For the Li|Cu cell with a PP-separator, it has sharp Li dendrites (Figure 4f) [63]. For the Li|Cu cell with either a BN-coated PP separator or an AlN-coated PP separator, the diameter of the formed Li dendrites is much larger than that in a Li|Cu cell with a PP separator [63,69]. This directly showed the influence of the thermally conductive material layer on the morphology of growing Li dendrites. However, most of previous works were focused on the intrinsic material properties of either BN or AlN, not on the practical side of coated layers. Considering this aspect, this topic should be further investigated by optimizing the morphology and size of coating materials and the thickness of the coating layer. It would eventually open a new avenue to use nitrides for designing functional separators compatible with practical LMBs.

### 3.2. Surface Functionalization of Polymer Separator with Metal Oxides

Metal oxides have common physical properties such as high melting/boiling point, strong mechanical strength, heat resistance, and corrosion resistance. As results of these attractive features, these are widely used in a variety of fields including LED, semiconductor, energy storage, etc. [126].

Battery industries such as LG Chemistry, Samsung SDI, etc. developed a safety-reinforced separator (SRS) (or termed ceramic-coated separator) using inorganic particles [127,128,129,130]. According to a few reports, SRS has an excellent mechanical/thermal stability and a superior electrolyte wettability, enabling the reliable and stable battery operation [127,128]. In this respect, introducing a metal oxide layer over the surface of the polymer separator would be a straightforward and efficient way to restrain the physical penetration of growing Li/Na dendrites and improve the thermal shrinkage problem of a polyolefin separator. There are a few available metallic oxides used for coating polymer separators. Cui et al. prepared a sandwich-structured separator comprising a SiO_2_ nanoparticle layer and two polyolefin separators (Figure 5a) [88]. The main concept is that the SiO_2_ nanoparticle layer between two polymer separators blocks the migration of growing Li dendrites by electrochemically consuming Li pillars passing through the punctured separator. This leads to an extended battery cycle life. However, sandwiching the SiO_2_ nanoparticle layer with two polymer separators is not an ideal approach for practical LMBs. In addition, the SiO_2_ nanoparticle layer has too many voids that Li dendrites are able to get through, which significantly increases the risk of dendrite penetration across the separator. Pol et al. proposed a SiO_2_ nanosheet-coated separator, instead of using SiO_2_ nanoparticles, in order to reduce the inter-particle space and make a compact and dense layer (Figure 5b) [67]. Interestingly, the separator with a SiO_2_ nanosheet layer enables a uniform Li deposition/stripping due to the nano-sized porosity of SiO_2_ nanosheets, leading to a stable cycle stability and a small increase in charge transfer resistance after multiple cycles. In addition, the designed separator showed the analogous results in Na metal batteries, which indicates that the SiO_2_ nanosheet is a promising coating material for developing practical separators compatible with alkaline metal-based batteries. Yuan et al. designed an Al_2_O_3_-separator via the doctor-blade method and tested it for Li symmetrical cells (Figure 5c) [116]. With the synergistic contribution of micro-sized alumina and nano-sized alumina, the thermal stability and electrolyte wettability of the prepared separator were greatly enhanced in comparison with a pristine PE separator, thus contributing to an improved cycle stability in Li symmetrical cells. Hu et al. fabricated an Al_2_O_3_-coated PVDF-HFP separator using an atomic layer deposition (ALD) process (Figure 5d) [117]. After Al_2_O_3_ coating, the material weaknesses such as low modulus and poor thermal stability of a pristine PVDF-HFP separator were greatly improved. Moreover, the electrolyte wettability (closely related to the ion conductivity) of the prepared separator was further enhanced in comparison with the pristine PVDF-HFP. Owing to these favorable characteristics, the electrochemical performances of Li|LiFePO_4_ cells and Li symmetrical cells were significantly ameliorated.

### 3.3. Surface Functionalization of Polymer Separator with Carbons

Carbons are widely used for a variety of applications, owing to its excellent electrochemical properties, mechanical strengths, and chemical stabilities [131,132]. In LIBs, carbons are typically employed as either active materials or additives in the electrode.

When seen from a different point of view, carbons can be applied as coating materials for fabricating functional separators for advanced LIBs. Pol et al. demonstrated a PDA/Gr-CMC separator designed for LMBs (Figure 6(a-1)) [66]. The surface of the PP separator was modified with polydopamine (PDA) followed by being coated with an aqueous graphene/CMC ink. The concept is to reduce the actual local current density applied to a cathode electrode by adding a lithiatable carbon layer over the PP separator. Note that Li is homogeneously deposited/stripped at low current densities, and sharp Li dendrites are prone to grow at high current densities [133,134]. The graphene layer participates in the electrochemical reaction by storing mobile Li^+^ inside the structure of graphene. The non-participated Li^+^ bypasses the separator and moves on to the cathode side. The Li|Cu cell with a PDA/Gr-CMC separator exhibited more stable cycle performance than the other control electrodes due to the presence of the graphene/CMC layer. After cycling, the tested cell was disassembled to observe the change of the graphene/CMC layer on the PDA/Gr-CMC separator (Figure 6(a-2)). It was revealed that mobile Li^+^ gets inside the graphene/CMC layer during the electrochemical tests, which significantly reduces the local current density of the counter electrode. In another study, Pol et al. used two pyrolyzed carbons derived from cellulose for fabricating electrodes and surface-modified separators (Figure 6(b-1)) [125]. Two cellulose nanomaterials, i.e., cellulose nanofibril (CNF) and cellulose nanocrystal (CNC), were employed as carbon precursors. One of the main differences between two nanocelluloses is that CNF has the amorphous section through the backbone of cellulose. After the pyrolysis process, each carbonized nanocellulose was laminated on the surface of the PP separator (Figure 6(b-2)). The Li|Cu cell with a c-CNF-separator delivered more stable cycle performance than the other two cells with different separators. It is mainly due to the different Li-storage capacity of each carbonized material. The c-CNF has a higher capacity than the c-CNC, which is due to the synergistic contribution of the amorphous region and crystalline region of CNF. It was ascertained that the carbon with higher Li-storage capacity is more beneficial to reducing the actual current density applied to the counter electrode in the cell. He et al. designed a VN@N-rGO-modified separator for Li metal-based batteries (Figure 6(c-1)) [110]. VN@N-rGO was prepared via hydrothermal treatment followed by an ammoniation process. The idea of this work is that the vanadium nitride (VN) nanosphere improves the ionic conductivity and electrolyte wettability of the formed layer on a PP separator. Reduced graphene oxide (rGO) offers an additional conducting path, suppresses the volume expansion, and reduces the overall local current density applied to the counter electrode (Figure 6(c-2)). With the synergistic contribution of each material, the cycle performance of Li symmetric batteries with the fabricated separator was dramatically improved. In addition, Li dendrite formation was effectively inhibited due to the presence of lithiatable graphene. Xie et al. suggested using functionalized nanocarbon (FNC) with immobilized Li ions for fabricating separators [111]. The role of the carbon layer on the fabricated separator is to control the Li dendrite growth orientation, not suppressing the formation of dendritic Li. This approach facilitates the transformation of the dendritic structure to a dense layer, which significantly reduces the risk of separator puncture and short. According to the electrochemical data of Li symmetrical cells, the designed separator exhibited a substantial effect on improving the Coulombic efficiency and cycle stability by controlling the growth orientation of Li dendrites. Concept-wise, it is an easy and effective approach to enable the long operation of the Li metal anode system with high efficiency. All these approaches of placing a carbon layer on the polymer separator have shown meaningful and promising outcomes in terms of controlling dendrite growth/orientation and improving the efficiency of Li deposition/dissolution.

However, the growth of dendritic Li is still unavoidable, even though the carbon layer on the separator can effectively suppress or change the feature of Li dendrites. In this regard, it would be eventually necessary to do broader research on either resolving the fundamental issue of Li dendrite growth itself or perfectly confining Li dendrites within the anodic side, using a hybrid composition of carbon and other functional materials.

### 3.4. Surface Functionalization of Polymer Separator with Metals

Metals have been widely used in many different fields due to their high electrical/thermal conductivity, malleability/ductility, high melting point and so on [135]. Especially in LIBs, metals are employed as either electrodes or current collectors.

Unlike the previous research studies, there have been different attempts to use metals as functional layers for stabilizing Li anodes. Song et al. suggested a new strategy to control Li dendrite growth by forming a metal (e.g., Mg) layer on one side of the polyolefin separator via sputtering (Figure 7a) [101]. The Mg layer with a lithiophilic nature efficiently reduces the Gibbs free energy (ΔG) for Li plating, thus leading to a uniform and dense Li deposition over the counter electrode. The electrochemical results directly corroborated that the lithiophilic Mg layer is able to improve the polarization and cycle stability of the cells having a Li metal anode. Huang et al. demonstrated a “dendrite-eating” separator by coating a Si layer over the PP separator (Figure 7b) [102]. The main role of the Si layer is to stabilize Li deposition and reduce the loss of available Li during the repetitive electrochemical reaction. According to the electrochemical data, the symmetrical cell with a Si-coated separator has a smaller polarization and a better cycle stability than the symmetrical cell with a pristine PP separator. This directly indicates the influence of the Si layer on improving the electrochemical performances of LMBs. Transparent symmetric batteries with and without a Si-coated separator were tested to monitor the morphological change of growing Li dendrites as a function of Li plating time. The cell with a Si-coated separator survived longer than the cell with a PP separator. It is because the Si coating not only plays a role in restraining the dendrite formation by alloying with Li dendrites but also distributes Li^+^ flux homogeneously across the separator. Zhang et al. proposed depositing an ultrathin Cu film on a PE separator for improving the electrochemical stability (Figure 7c) [99]. The ultrathin Cu film can change the growth orientation of Li dendrites from the vertical direction to the horizontal direction as well as diminish the local current density by exposing more surface area of the Li layer. Due to these beneficial functions of the ultrathin Cu layer, the cycle performance and Coulombic efficiency of Li|Cu cells were dramatically ameliorated in comparison with the Li|Cu cell with a pristine PE separator. Murugan et al. reported a binder-free Nb-coated PP separator for LMBs (Figure 7d) [100]. The Nb layer alters the surface nature of the PP separator from hydrophobicity to hydrophilicity, contributing to an improved electrolyte wettability. In addition, the formed metal layer not only enhances the mechanical strength of the conventional PP separator but also serves as an additional conductive path inducing the merge of Li dendrites from both sides of the anode and Nb layer. Moreover, the improved contact between the Nb layer and Li anode and the homogeneous current flow efficiently reduces the interfacial resistance between the anode and electrolyte. All these advantageous features greatly mitigate the risk of a short circuit and improve the cycle life with small polarization in LMBs.

### 3.5. Surface Functionalization of Polymer Separator with Polymers

Polymers are extensively used as binding materials, separators, and electrolytes in rechargeable batteries due to the high mechanical stability, the excellent chemical resistance, and the good adhesion properties [136]. In addition to these purposes, polymers are also adopted as coating materials for separator modification.

Han et al. fabricated a polyimide (PI)-coated PE separator for LMBs (Figure 8a) [103]. The PE separator was modified with PI microspheres and PI nanofibers via the electrospinning method. When the prepared PI–PE separator was introduced in Li symmetrical cells, it dramatically improved the cycle stability without degraded Coulombic efficiency. It is because the PI layer improves the electrolyte wettability and enables the uniform Li-ion transfer through the Li^+^-rich channel between the Li anode and the PI layer. In addition, the PI coating significantly improves the thermal property of the PE separator up to 120 °C, which is confirmed by the shrinkage test at elevated temperature. Choi et al. employed polydopamine coating to typical PE separators, in order to modify the surface property of separators from hydrophobicity to hydrophilicity (Figure 8b) [104]. After polydopamine coating, the color of the white PE separator turned brown, and the porosity of the PE separator did not change. The surface-modified separator plays a role in improving the electrolyte wettability as well as homogenizing Li^+^ flux. The enhanced adhesion property, which is attributed to a catecholic interaction, between the separator and Li anode efficiently releases the local surface tension of the Li metal anode, resulting in the formation of flat flake-like Li dendrites, not the formation of sharp Li dendrites. It is also reported that the polydopamine layer can mitigate the detrimental chemical crossover and enhance the electrode stability [70]. All these favorable features contributed to the improved electrochemical stability of Li metal anodes. Zhang et al. formed a lithiophilic polymer layer comprising chitosan, poly(ethylene oxide) (PEO), and triethylene glycol dimethacrylate (TEGDMA) monomers over the Celgard separator via the electrospraying and polymerization process (Figure 8c) [105]. The synthesized layer enables uniform Li deposition on the anode, serves as an artificial SEI layer, and blocks the penetration of growing Li dendrites across the separator. With the aid of the lithiophilic polymer layer, the cycle retention and rate performance of Li|Cu cells and Li|LFP full cells were remarkably enhanced. Yuan et al. tuned the surface of the PE separator with polydopamine/octaammonium POSS (PDA/POSS) via a dip-coating process (Figure 8d) [106]. The ultrathin polymer layer not only affects the ionic conductivity and Li^+^ transference number but also improves the interface stability between Li and the electrolyte. These electrochemically favorable effects significantly reduce the electrode polarization and curb the dendritic Li growth, contributing to the improved electrochemical performances of LMBs. Similarly, Yuan et al. optimized the PE-separator surface using multilayer PEI(PAA/PEO)_3_ via a layer-by-layer (LBL) assembly process (Figure 8e). The role of the polymer layer is to improve the ionic conductivity of the separator and stabilize the surface of the Li anode against liquid electrolyte. Due to these effects, the long-term cycle stability of LMBs is achieved while maintaining high Coulombic efficiency [107]. Li et al. proposed a process of using C-N polymer to form N-deficient thin film (Figure 8f) [108]. The formed film not only plays a role as an artificial SEI layer but also homogenizes Li plating over the counter electrode. The prepared N-rich C-N powder was coated onto a PP separator via tape casting, and the fabricated separator was used for electrochemical tests. According to the cycle performance (Coulombic efficiency vs. cycle number), the Li|Cu cell with a g-C_3_N_4_ modified separator delivered a more stable cycle retention with higher Coulombic efficiency than the control cell due to the electrochemically favorable role of the C_3_N_4−x_ layer.

### 3.6. Surface Functionalization of Polymer Separator with Other Materials (such as Solid-State Electrolytes)

Typically, solid-state electrolytes are prepared as pellets for the battery assembly. Different from the previous approach, there is another attempt to use solid-state electrolytes as separator-coating materials. For instance, Sun et al. designed a hybrid separator using Li_6.75_La_3_Zr_1.75_Ta_0.25_O_12_ (LLZTO) and aramid nanofibers via a simple casting method (Figure 9a) [120]. The prepared separator possesses an excellent mechanical stability as well as high thermal stability. With these beneficial advantages, the fabricated separator can maintain its structural integrity even at a high temperature around 150 °C. In addition, the LLZTO layer is able to distribute Li ions back and forth uniformly, consequently leading to the reduced formation of dendritic Li. The Li symmetrical cell with the hybrid separator exhibited a highly stable cycle performance without significant polarization fluctuation as opposed to other two control cells. Zhang et al. deposited a solid-state electrolyte layer, comprising Al-doped Li_6.75_La_3_Zr_1.75_Ta_0.25_O_12_ (LLZTO) and PVDF, on one side of the PP separator (Figure 9b) [121]. The fabricated separator can be fully bended without noticeable delamination. The finite element method (FEM) simulation proved the substantive effect of the designed separator on redistributing Li ions through the 3D channel of the LLZTO layer. Owing to these beneficial effects, the cell with the modified separator exhibited a longer cycle life (over 75 h) and showed a smaller potential polarization than the cell with the PP separator.

A similar concept has been studied by introducing a layer comprising of Li_6.4_La_3_Zr_1.4_Ta_0.6_O_12_ (LLZTO) and PVDF on the PP separator (Figure 9c) [122]. The prepared separator not only regulates Li^+^ flux but also immobilizes anions during the electrochemical reactions. With the aid of the synergistic contribution of LLZTO and PVDF, the Coulombic efficiency and cycle stability of LMBs have been significantly improved, along with small voltage polarization. To sum up, all the approaches using the composites of solid-state electrolytes and polymeric binders have shown promising results in controlling the formation of dendritic Li and extending the life span of Li metal anodes.

## 4. Conclusions and Outlooks

Typical polyolefin separators are not physically robust enough to restrain the propagation of growing Li/Na dendrites, which allows the unavoidable penetration of dendritic Li through the separator. In addition, polyolefin separators are not thermally stable to maintain its structural integrity at high temperatures and have relatively poor electrolyte wettability. These material limits hinder the practical use of polymer separators for high-performance batteries with Li metal anode. The surface modification of polymer separators with functional materials can play an important role in homogenizing Li^+^ flux, preventing a physical penetration of alkaline metal dendrite, and strengthening a thermal/mechanical stability of separators. This paper reviews the characteristics and limits of existing battery separators and summarizes the overall separator-coating materials that are effective in controlling the growth of dendritic Li and improving the efficiency of reversible Li^+^ transport. Surface-functionalized separators designed for LMBs can be classified into five or six categories according to the types of separator coating materials: (a) thermally conductive materials, (b) metal oxides, (c) carbons, (d) polymers, (e) metals, and (f) others (including solid-state electrolytes). Each type of material has different material properties, directly leading to different electrochemical results when applied to the surface of polymer separators for LMBs.

To date, there has been meaningful progress in fabricating functional separators for next-generation batteries including Li-S batteries and Li/Na metal batteries. With the adoption of the customized separators, the battery’s electrochemical/thermal stability has been remarkably improved in comparison to the existing battery technologies. These technologies are currently used by global battery industries (e.g., LG Chem., Samsung SDI, SK Innovation, etc.). Unfortunately, further improvements are still required to use the surface-modified separator over the long term and reduce the side reactions that mostly happen inside the Li metal cell. For example, it is no longer possible to homogenize Li^+^ flux and block the Li dendrite penetration once the coating layer is fully covered by metallic Li during the electrochemical test. It is because the Li-deposited layer itself can serve as another Li anode inside the cell.

Considering the potential problems, there are a few parts where researchers and engineers can be devoted to upgrading the previous approaches. For instance, constructing a hybrid film comprising more than two types of separator-coating materials (described in Figure 3) on the polymer separator might be a feasible strategy to have multiple synergistic effects on improving the reversibility and stability of LMBs. The combination of a gel electrolyte with a surface-functionalized separator would be another approach to address the interface issues such as a non-uniform contact between a coating layer and an electrode. Alongside these suggested ideas, there are many niche markets where researchers can contribute to the further improvement of battery separators for practical LMBs. In addition to these conceptual approaches, the cost reduction of manufacturing functional separators is another important factor to be considered for successful commercialization. Since most of the previous studies were mainly focused on fabricating the surface-functionalized separator on a laboratory scale (using either tape-casting or sputtering), it would significantly increase the manufacturing cost of functional separators compatible with the battery assembly process.

In this regard, there are still many hurdles for researchers to go through. The topic of surface-functionalized separators needs more relevant and meaningful research to overcome the inherent weaknesses of polymer separators and open a new route to customize the properties of polymer separators.

## Figures and Tables

**Figure 1 nanomaterials-11-02275-f001:**
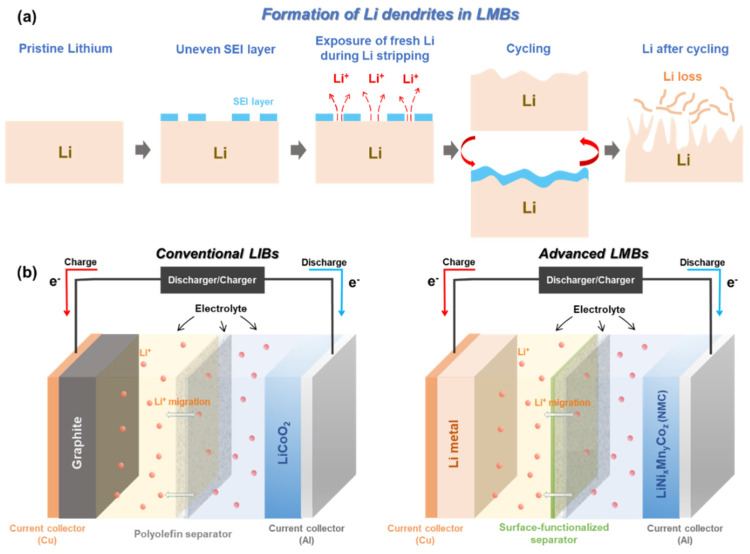
(**a**) Schematics to illustrate how the surface of Li metal anode changes during electrochemical reaction. (**b**) Configurations of a conventional LIB introduced with a polyolefin separator and an advanced LMB introduced with a surface-functionalized separator.

**Figure 2 nanomaterials-11-02275-f002:**
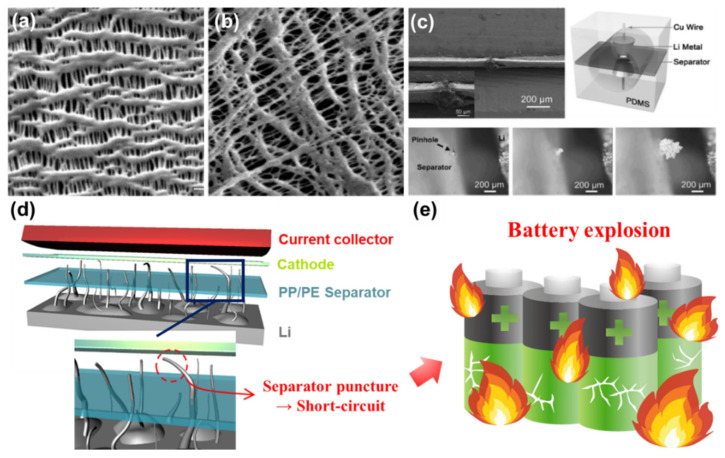
Top-view SEM images of separators made by (**a**) a dry method and (**b**) a wet method. (Reproduced with permission from [73]. Copyright 2004 American Chemical Society) (**c**) In-situ transparent cell test to show how Li dendrite grows through the pinhole of the separator. This finding indicates that an internal short-circuit occurs by mossy Li dendrites. (Reproduced with permission from [88]. Copyright 2016 WILEY-VCH) (**d**) Schematics to illustrate how Li dendrite pierces the polyolefin separator and consequently causes a battery short-circuit. (Reproduced with permission from [67]. Copyright 2018 Elsevier) (**e**) The short-circuit results in a serious safety problem such as battery explosion.

**Figure 3 nanomaterials-11-02275-f003:**
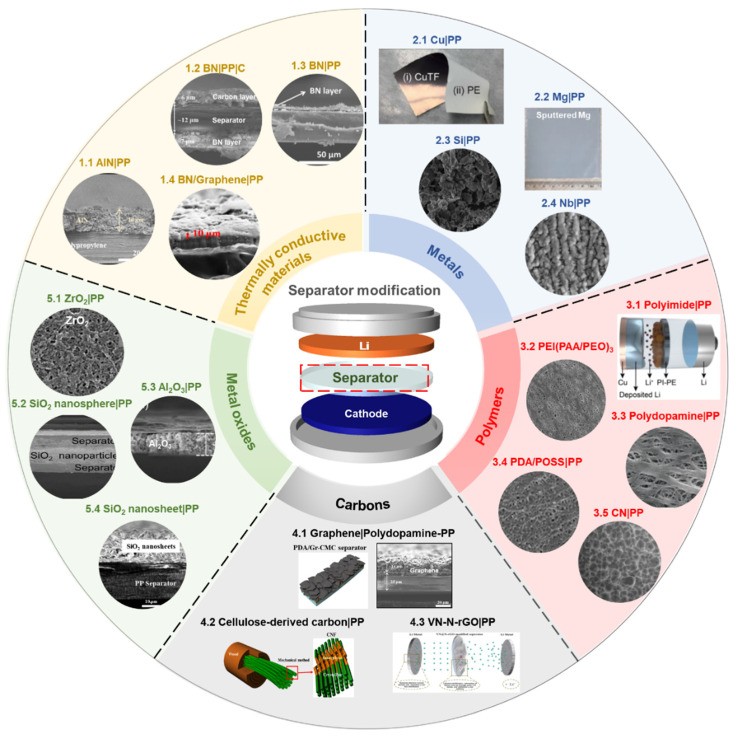
Thermally conductive materials: 1.1 AlN (Reproduced with permission from [63], Copyright 2019 American Chemical Society), 1.2 BN&C (Reproduced with permission from [64], Copyright 2017 Springer Nature), 1.3 BN (Reproduced with permission from [69], Copyright 2015 American Chemical Society), 1.4 BN/Graphene (Reproduced with permission from [97], Copyright 2020 Elsevier). Metals: 2.1 Cu (Reproduced with permission from [99], Copyright 2017 WILEY-VCH), 2.2 Mg (Reproduced with permission from [101], Copyright 2018 Elsevier), 2.3 Si (Reproduced with permission from [102], Copyright 2020 Elsevier B.V.), 2.4 Nb (Reproduced with permission from [100], Copyright 2019 Springer Nature). Polymers: 3.1 Polyimide (Reproduced with permission from [103], Copyright 2021 American Chemical Society), 3.2 PEI(PAA/PEO)_3_ (Reproduced with permission from [107], Copyright 2018 American Chemical Society), 3.3 Polydopamine (Reproduced with permission from [104], Copyright 2012 WILEY-VCH), 3.4 PDA/POSS (Reproduced with permission from [106], Copyright 2017 Elsevier), 3.5 CN (Reproduced with permission from [108], Copyright 2020 American Chemical Society). Carbons: 4.1 Graphene/Polydopamine (Reproduced with permission from [66], Copyright 2018 WILEY-VCH), 4.2 Cellulose-derived carbon (Reproduced with permission from [125], Copyright 2020 Elsevier), 4.3 VN-N-rGO (Reproduced with permission from [110], Copyright 2021 American Chemical Society). Metal oxides: 5.1 ZrO_2_ (Reproduced with permission from [115], Copyright 2016Elsevier), 5.2 SiO_2_ (Reproduced with permission from [88], Copyright 2016 WILEY-VCH), 5.3 Al_2_O_3_ (Reproduced with permission from [116], Copyright 2020 WILEY-VCH), 5.4 SiO_2_ (Reproduced with permission from [67], Copyright 2018 Elsevier).

**Figure 4 nanomaterials-11-02275-f004:**
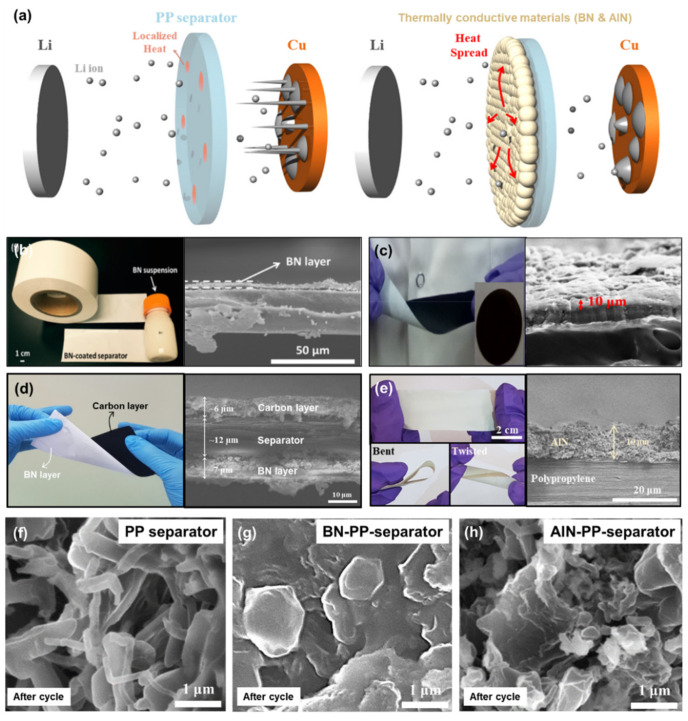
Thermally conductive materials. (**a**) Schematics to illustrate how a thermally conductive material layer spreads the localized heat inside LMBs and affects the morphology of Li dendrite. (Reproduced with permission from [63], Copyright 2019 American Chemical Society) Photos and SEM images of (**b**) a BN-coated separator (Reproduced with permission from [69], Copyright 2015 American Chemical Society), (**c**) a BN-graphene-coated separator (Reproduced with permission from [97], Copyright 2020 Elsevier), (**d**) a tri-layered separator (BN|PP-separator|Carbon) (Reproduced with permission from [64], Copyright 2017 Springer Nature) and (**e**) an AlN-coated separator (Reproduced with permission from [63], Copyright 2019 American Chemical Society). SEM images of Li anode surface in the cycled Li|Cu cells with (**f**) a PP-separator (Reproduced with permission from [63], Copyright 2019 American Chemical Society), (**g**) a BN-PP-separator (Reproduced with permission from [64], Copyright 2017 Springer Nature) and (**h**) an AlN-PP-separator (Reproduced with permission from [63], Copyright 2019 American Chemical Society).

**Figure 5 nanomaterials-11-02275-f005:**
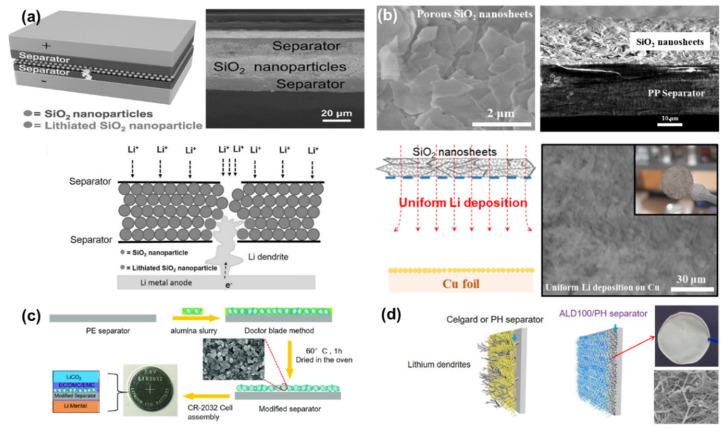
Metal oxides. (**a**) The structure of a sandwich-structured SiO_2_-separator for LMB evaluation and a schematic to show how Li dendrite penetration is inhibited by SiO_2_ nanoparticles. (Reproduced with permission from [88], Copyright 2016 WILEY-VCH) The area where Li dendrite contacts SiO_2_ nanoparticle is electrochemically reacted to become lithiated SiO_2_. (**b**) The morphology of a SiO_2_ nanosheet-coated separator and a schematic to illustrate how a SiO_2_ nanosheet layer plays a role in homogenizing the Li deposition. As shown in the picture and SEM image, Li is uniformly deposited over the Cu foil. (Reproduced with permission from [67], Copyright 2018 Elsevier) (**c**) Schematics to illustrate the steps of preparing an Al_2_O_3_-coated separator by tape-casting method. (Reproduced with permission from [116], Copyright 2020 WILEY-VCH) (**d**) Schematics to compare a pristine separator and an Al_2_O_3_-coated PVDF-HFP separator prepared by ALD. (Reproduced with permission from [117], Copyright 2019 American Chemical Society).

**Figure 6 nanomaterials-11-02275-f006:**
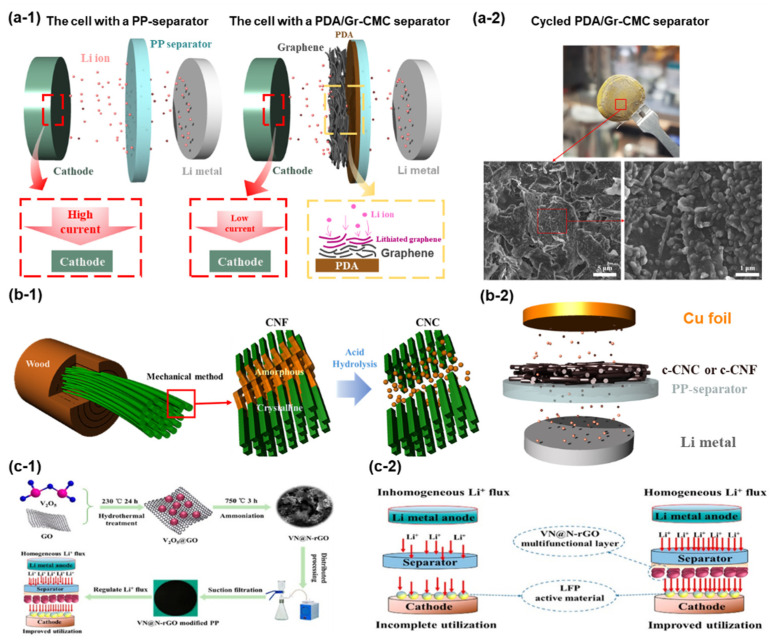
Carbons. (**a-1**) Schematics to illustrate the role of a graphene layer on reducing the local current density applied to the cathode. (**a-2**) SEM images of cycled separators. (Reproduced with permission from [66], Copyright 2018 WILEY-VCH) (**b-1**) Schematics to show the difference of CNC and CNF. (**b-2**) Configuration of a Li|Cu cell with either c-CNC or c-CNF-coated separator. (Reproduced with permission from [125], Copyright 2020 Elsevier) (**c-1**) Preparation step of VN@N-rGO modified PP separator. (**c-2**) Role of VN@N-rGO layer in LMBs. (Reproduced with permission from [110], Copyright 2021 American Chemical Society).

**Figure 7 nanomaterials-11-02275-f007:**
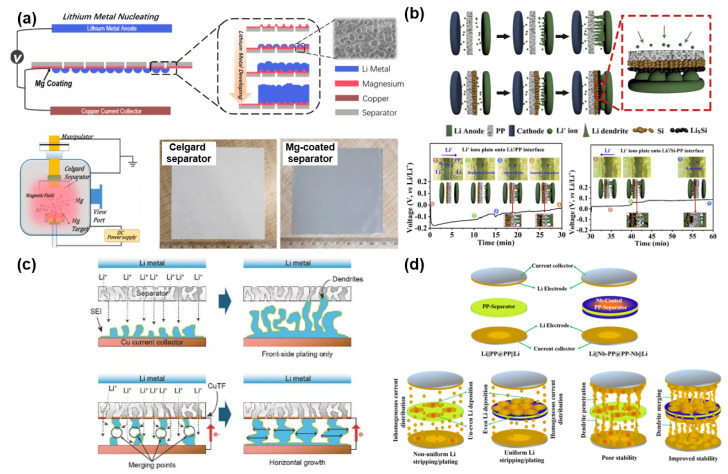
Metals. (**a**) Schematics to illustrate the preparation step of Mg-coated separator and its influence on Li metal anode. (Reproduced with permission from [101], Copyright 2018 Elsevier) (**b**) Schematics to illustrate the influence of Si layer on Li dendrite growth. (Reproduced with permission from [102], Copyright 2020 Elsevier B.V.) (**c**) Schematics to illustrate the role of Cu thin film on Li dendrite growth. (Reproduced with permission from [99], Copyright 2017 WILEY-VCH) (**d**) Comparison between PP-separator and Nb-coated PP-separator. (Reproduced with permission from [100], Copyright 2019 Springer Nature).

**Figure 8 nanomaterials-11-02275-f008:**
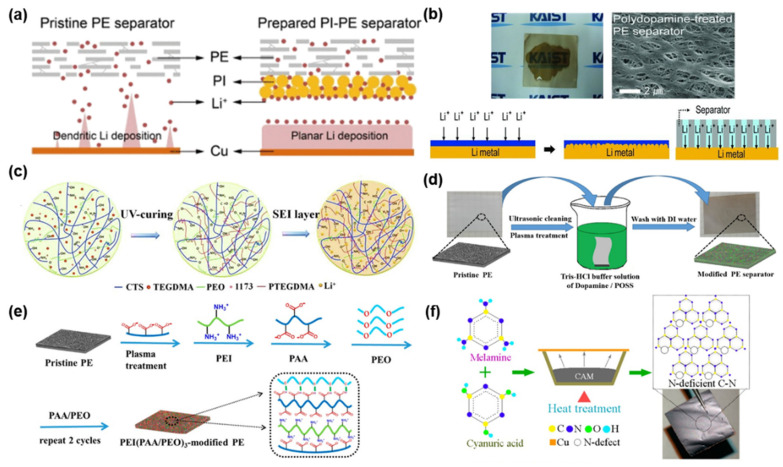
Polymers. (**a**) Schematics to illustrate the roles of PI microsphere layer on Li deposition. (Reproduced with permission from [103], Copyright 2021 American Chemical Society) (**b**) Images of polydopmine-coated PE-separator and the effect of the polydopamine layer on Li^+^ flux and Li deposition. (Reproduced with permission from [104], Copyright 2012 WILEY-VCH) (**c**) Schematics to show the formation of a crosslinked 3D s-IP CTS-PEO-PTEGDMA layer. (Reproduced with permission from [105], Copyright 2020 Elsevier B.V.) (**d**) Fabrication of PDA/POSS modified-PE separators. (Reproduced with permission from [106], Copyright 2017 Elsevier) (**e**) Preparation of PEI(PAA/PEO)_3_-modified PE-Separator. (Reproduced with permission from [107], Copyright 2018 American Chemical Society) (**f**) Preparation of N-deficient C-N films. (Reproduced with permission from [108], Copyright 2020 American Chemical Society).

**Figure 9 nanomaterials-11-02275-f009:**
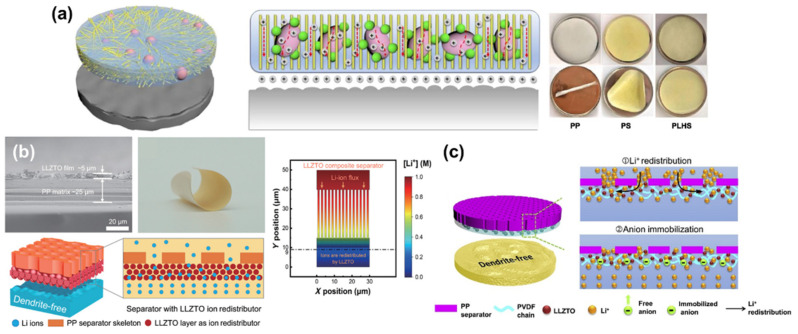
Others. (**a**) Schematics to show the benefits of PLHS-coated separators in terms of thermal shrinkage and electrochemical performance. (Reproduced with permission from [120], Copyright 2021 Elsevier B.V.) (**b**) The structure and electrochemical benefits of LLZTO-coated separator for LMBs. The distribution of Li-ion concentration is modeled as a graph. (Reproduced with permission from [121], Copyright 2018 American Association for the Advancement of Science) (**c**) Schematics to illustrate how PP@PLLZ-separator enables the homogeneous Li^+^ flux. (Reproduced with permission from [122], Copyright 2019 Elsevier B.V.).

**Table 1 nanomaterials-11-02275-t001:** Summary of separator-coating materials according to material categories.

Separator-Coating Materials	Separator & Fabrication Technique	Coulombic Efficiency, Current Density, Tested Cells	Ref.
**1. Thermally conductive materials**			
AlN nanopowder	PP, Tape-casting	92% at 1st and 92% at 100th cycle, 0.5 mA cm^−2^, Li|Cu	[63]
BN nanopowder&carbon	PP, Tape-casting	90% at 1st and 83.5% at 100th cycle, 0.5 mA cm^−2^, Li|Cu	[64]
BN nanosheets	PP/PE/PP, Spray coating	88% at 1st and 92% at 100th cycle, 0.5 mA cm^−2^, Li|Cu	[69]
BN nanopowder/Graphene	PP, Tape-casting	77% at 1st and 75% at 100th cycle, 1 mA cm^−2^, Li|Cu	[97]
**2. Metals**			
Cu	PE, Magnetron sputtering	95.4% at 1st and 95.6% at 300th cycle, 0.4 mA cm^−2^, Li|Cu	[99]
Nb	PP, Magnetron sputtering	97% at 120th cycle, 0.2 C, Li|LNMC	[100]
Mg	PP/PE, Magnetron sputtering	97% at 1st and 94% at 400th cycle, 0.5 mA cm^−2^, Li|Cu	[101]
Si	PP, Tape-casting	90% at 1st and 97.6% at 100th cycle, 0.5 mA cm^−2^, Li|Cu	[102]
**3. Polymers**			
Polyimide	PE, Electrospinning	98.5% at 700th cycle, 1 mA cm^−2^, Li|Cu	[103]
Polydopamine	PE, Solution immersion	97.1% at 1st cycle, 0.17 mA cm^−2^, Li|LiCoO_2_	[104]
CTS-PEO-PTEGDMA	PP/PE/PP, Electrospraying	~60% at 1st and 90% at 120th cycle, 0.5 mA cm^−2^, Li|Cu	[105]
PDA/POSS	PE, Dip coating	98.6% at 200th cycle, 0.2 C, Li|LiCoO_2_	[106]
PEI(PAA/PEO)_3_	PE, Layer-by-layer (LBL)	99.1% at 400th cycle, 0.2 C, Li|LiCoO_2_	[107]
C-N polymer	PP, Pasting	~72% at 1st and 95% at 450th cycle, 3 mA cm^−2^, Li|Cu	[108]
**4. Carbons**			
Graphene|Polydopamine	PP, Tape-casting	~96.1% at 1st and 86.6% at 200th cycle, 0.5 mA cm^−2^, Li|Cu	[66]
VN-N-rGO	PP, Vacuum filtration	92% at 1st and 97.1% at 55th cycle, 0.5 mA cm^−2^, Li|Cu	[110]
Functionalized nanocarbon	PP, Tape-casting	96.07–97.41%, 1 mA cm^−2^, Li|Li	[111]
Cellulose-derived carbon	PP, Tape-casting	36.7% at 1st and 77.1% at 120th cycle, 1 mA cm^−2^, Li|Cu	[125]
**5. Metal oxides**			
SiO_2_ nanosheet	PP, Tape-casting	81.5% at 1st and 64% at 200th cycle, 1 mA cm^−2^, Li|Cu	[67]
SiO_2_ nanoparticles	PE, Tape-casting	-	[88]
ZrO_2_	PE, Self-assembly	97.8% at 400th cycle, 0.5 C, Li|LiCoO_2_	[115]
Al_2_O_3_ particles	PE, Tape-casting	nearly 100% at 100th cycle, 0.2 C, Li|LiCoO_2_	[116]
Al_2_O_3_	PVDF-HFP, ALD	nearly 100% at 100th cycle, 0.2 C, Li|LiFePO_4_	[117]
**6. Others**			
Li_6.75_La_3_Zr_1.75_Ta_0.25_O_12_	PP, Tape-casting	99.5% after 1000 cycles, 0.2 C, Li|LiFePO_4_	[120]
Al-doped Li_6.75_La_3_Zr_1.75_Ta_0.25_O_12_	PP, Vacuum filtration	98% after 450 cycles, 0.5 mA cm^−2^, Li|Cu	[121]
Li_6.4_La_3_Zr_1.4_Ta_0.6_O_12_	PP, Tape-casting	97.5% after 300 cycles, 1 mA cm^−2^, Li|Cu	[122]

## Data Availability

Not applicable.

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
