# Peer review of "Surface-Functionalized Separator for Stable and Reliable Lithium Metal Batteries: A Review"

_nanomaterials, 2021, doi:10.3390/nano11092275_

Round 1

Reviewer 1 Report

The review “Surface-Functionalized Separator for Stable and Reliable Lithium Metal Batteries: a review” describes the role of functionalized separator in Li-ions batteries. The article can be considered for publication in Nanomaterials after major revision. Here the comments:

  • As first the author must consider the commercial aspects: how does functionalization affect the costs? In conclusions and outlook part, the authors should propose new perspectives for industry. What is the most promising commercial trend?
  • The authors should add a table (or an image) at the end of the paper showing what is (are) the best separator(s) in full battery with a) highest loading, b) highest current densities applied and c)best . Probably the author should consider NMC, LFP and sulfur cathode chemistries. This table (or image) can help to design future research works.
  • About the use of carbon as interlayer: is it possible a loss of voltage having consequently a lower energy density?
  • The authors have to add few sentences in the introduction about the evolution of Li-ion batteries and their role in our society: a) Journal of Solid State Electrochemistry 15, 2011, 1623–1630; b) Materials 13 (8), 2020, 1884; c) b) Angew. Chem. Int. Ed. 51, 2012, 5798 – 5800;
  • About Lithium sulfur batteries some important reports are missing: a) ACS Appl. Mater. Interfaces 11, 2019, 11459–11465; b) Nano Energy 30, 2016, 1–8 c) Journal of Power Sources 427, 2019, 201-206; d) Electrochimica Acta 192, 2016, 346–356  e) The Journal of Physical Chemistry C 122 (2), 2018, 1014-1023; f) ACS Appl. Energy Mater. 2, 2019, 5793–5798
  • The authors should consider the loss of capacity is due to a slow diffusion of organic compounds inside the electrolyte during cycling and the role played by the separator to address this issue, therefore the following papers related to this subject must be cited: a) ACS Energy Lett. 2018, 3, 12, 2921–2930 b) Scientific Reports 9, 2019, 1213;  c) Journal of Materials Chemistry A, 5, 2017, 6532–6537
  • The authors must consider other articles about flame retardants such as: a) DOI: 10.1126/sciadv.1601978; b) Journal of Power Sources 506, 2021, 230189; c) Journal of Materials Chemistry A, 8, 2020, 14788-14798; d) Journal of Energy Chemistry 64, 2022, 372–384; e) Angewandte Chemie 131, 2019, 7884 –7889
  • The authors must add a discussion about the influence of electrolyte on battery performance. Some reports must be cited: a) Advanced Materials 32, 2020, 1904205; b) Materials Science and Engineering: R: Reports 134, 2018, 1-21; c) Journal of Electrochemical Society 161, 2014 , A1001
  • The importance of SEI nature should be discussed more in details: a) Journal of Power Sources 377, 2018, 7–11; b) Electrochimica Acta 51, 2006, 1636–1640 c) Communications Chemistry 2, 2019, Article number: 131; d) ACS Applied Materials Interfaces 12, 2020, 33719–33728

Author Response

Editor and reviewer comments

Reviewer: 1

Recommendation: Major revisions needed as noted.

Overall Assessment: The review “Surface-Functionalized Separator for Stable and Reliable Lithium Metal Batteries: a review” describes the role of functionalized separator in Li-ions batteries. The article can be considered for publication in Nanomaterials after major revision. Here the comments:

Our response: We thank the reviewer for the overall assessment on our manuscript.

Comment 1. As first the author must consider the commercial aspects: how does functionalization affect the costs? In conclusions and outlook part, the authors should propose new perspectives for industry. What is the most promising commercial trend?

Our response: We appreciate the fruitful comment from the reviewer. As reviewer commented, we edit the manuscript in the conclusion and outlook part. Changed parts are highlighted.

Comment 2. The authors should add a table (or an image) at the end of the paper showing what is (are) the best separator(s) in full battery with a) highest loading, b) highest current densities applied and c)best . Probably the author should consider NMC, LFP and sulfur cathode chemistries. This table (or image) can help to design future research works.

Our response: We appreciate the fruitful comment from the reviewer. The main storyline of this paper is to show how the separator coating layer can affect the cyclability and Coulombic efficiency of Li metal anode. Each coating material has different electrochemical properties when it was applied to polyolefin separators. Considering this fact, it is hard to tell which separator is the best. But, the objective of this article is to give an overview of separator coating materials in order for readers to refer to the characteristics of coating materials. In case of the chemistry of Li-S batteries, it is relatively complicated and different from Li|Cu cells and symmetric Li cells. So, we have mainly focused on the Coulombic efficiency of either symmetric Li cells and Li|Cu cells at a certain cycle number for fair comparison. In many cases, loading mass was not provided in the article, since authors want to compare their separator with conventional polyolefin separator. Therefore, we try to compare the cycle data of Li | Cathode full cell, instead. I believe you understand our intention.

Comment 3. About the use of carbon as interlayer: is it possible a loss of voltage having consequently a lower energy density?

Our response: We thank the reviewer’s comment. There are two important roles of the carbon layer on the polyolefin separator when it is applied to Li metal batteries. 1) It reduces the actual local current densities applied to counter electrode. 2) There is a chance that the carbon layer changes the growth orientation of Li dendrite. We put the relevant information in the main text. (NATURE ENERGY 2, 17083 (2017)) According to the article (Advanced Energy Materials 8 (36), 1802665), the full cell (Li vs. LFP) with the graphene-modified separator has shown improved cycle stability due to the above-mentioned effects of the carbon layer. In this regard, it would be okay to mention that the carbon layer does not affect the loss of voltage. Thanks for your fruitful comment.

Comment 4. The authors have to add few sentences in the introduction about the evolution of Li-ion batteries and their role in our society: a) Journal of Solid State Electrochemistry 15, 2011, 1623–1630; b) Materials 13 (8), 2020, 1884; c) b) Angew. Chem. Int. Ed. 51, 2012, 5798 – 5800

Our response: We appreciate the comment from the reviewer. We have edited the main-text and added all the references in the main-text based upon the comment. Thanks.

Comment 5. About Lithium sulfur batteries some important reports are missing: a) ACS Appl. Mater. Interfaces 11, 2019, 11459–11465; b) Nano Energy 30, 2016, 1–8 c); d) Electrochimica Acta 192, 2016, 346–356  e) The Journal of Physical Chemistry C 122 (2), 2018, 1014-1023; f) ACS Appl. Energy Mater. 2, 2019, 5793–5798

Our response: We appreciate the comment from the reviewer. We have edited the main-text and added all the references in the main-text based upon the comment. Thanks.

Comment 6. The authors should consider the loss of capacity is due to a slow diffusion of organic compounds inside the electrolyte during cycling and the role played by the separator to address this issue, therefore the following papers related to this subject must be cited: a) ACS Energy Lett. 2018, 3, 12, 2921–2930 b) Scientific Reports 9, 2019, 1213; c) Journal of Materials Chemistry A, 5, 2017, 6532–6537

Our response: We appreciate the comment from the reviewer. We have edited the main-text and added all the references in the main-text based upon the comment. Thanks.

Comment 7. The authors must consider other articles about flame retardants such as: a) DOI: 10.1126/sciadv.1601978; b) Journal of Power Sources 506, 2021, 230189; c) Journal of Materials Chemistry A, 8, 2020, 14788-14798; d) Journal of Energy Chemistry 64, 2022, 372–384; e) Angewandte Chemie 131, 2019, 7884 –7889

Our response: We appreciate the comment from the reviewer. We have edited the main-text and added all the references in the main-text based upon the comment. Thanks.

Comment 8. The authors must add a discussion about the influence of electrolyte on battery performance. Some reports must be cited: a) Advanced Materials 32, 2020, 1904205; b) Materials Science and Engineering: R: Reports 134, 2018, 1-21; c) Journal of Electrochemical Society 161, 2014 , A1001

Our response: We appreciate the comment from the reviewer. We have edited the main-text and added all the references in the main-text based upon the comment. Thanks.

Comment 9. The importance of SEI nature should be discussed more in details: a) Journal of Power Sources 377, 2018, 7–11; b) Electrochimica Acta 51, 2006, 1636–1640 c) Communications Chemistry 2, 2019, Article number: 131; d) ACS Applied Materials Interfaces 12, 2020, 33719–33728

Our response: We appreciate the comment from the reviewer. We have edited the main-text and added all the references in the main-text based upon the comment. Thanks.

Reviewer 2 Report

In this paper, separator-coating materials are classified into six categories to give a general guideline for fabricating functional separators compatible with post lithium-ion-batteries. The overall research trends and outlook for surface-functionalized separators
are reviewed. This work is written well and orgnized reasonably. I wonder recommend it for publication in nanomaterials after some minor revisons.

  1. In the lines 76-77, the aushor pointed out that "All of these unwanted phenomena become even more serious when LMBs are operating under abuse conditions, i.e. high/low temperature, high current density and overcharging". More discussion should be provided on the temperatures dependency for LMBs, such as: International Journal of Plasticity 88 (2017) 188-203; Philosophical Magazine 99 (2019), 992-1013; Journal of Materials Science 53 (2018), 10987-11001.
  2. How to juge the the total thickness of the separator is appropriate?  Please give more physical explanations.
  3. Surface functionalization seems to be a very good method to prepare separator materials. It is better if the authors could analyze some interface bonding properties of separator itself and other surrouding materials by tension, bending, and XPS analysis, such as: Extreme Mechanics Letters 9 (2016) 226-236; Journal of The Electrochemical Society 163 (2016) A1157-A1163; Journal of Power Sources 290 (2015) 114-122; ACS Applied Materials & Interfaces 11 (2019) 24648–24658.

Author Response

Editor and reviewer comments

Reviewer: 2

Recommendation: Minor revisions needed as noted.

Overall Assessment: In this paper, separator-coating materials are classified into six categories to give a general guideline for fabricating functional separators compatible with post lithium-ion-batteries. The overall research trends and outlook for surface-functionalized separators are reviewed. This work is written well and orgnized reasonably. I wonder recommend it for publication in nanomaterials after some minor revisons.

Our response: We thank the reviewer for the overall assessment on our manuscript.

Comment 1. In the lines 76-77, the aushor pointed out that "All of these unwanted phenomena become even more serious when LMBs are operating under abuse conditions, i.e. high/low temperature, high current density and overcharging". More discussion should be provided on the temperatures dependency for LMBs, such as: International Journal of Plasticity 88 (2017) 188-203; Philosophical Magazine 99 (2019), 992-1013; Journal of Materials Science 53 (2018), 10987-11001.

Our response: We appreciate the comment from the reviewer. We have added all the references in the article based upon the comment. Thanks.

Comment 2. How to juge the the total thickness of the separator is appropriate?  Please give more physical explanations.

Our response: We appreciate the fruitful comment from the reviewer. There is always a chance that the coating layer increases the overall impedance of the cell. Therefore, it is desirable to reduce the thickness of the coating layer as possible as we can. But in some case, if the coating layer is too thin, it would rather aggravate the electrochemical performances of Li metal batteries. In this regard, it itself is another topic to optimize the thickness of the coating layer and tune the size of coating materials for achieving stable Li metal batteries, which is described in the main-text. We highly appreciate your suggestion and valuable comments. 

Comment 3. Surface functionalization seems to be a very good method to prepare separator materials. It is better if the authors could analyze some interface bonding properties of separator itself and other surrouding materials by tension, bending, and XPS analysis, such as: Extreme Mechanics Letters 9 (2016) 226-236; Journal of The Electrochemical Society 163 (2016) A1157-A1163; Journal of Power Sources 290 (2015) 114-122; ACS Applied Materials & Interfaces 11 (2019) 24648–24658.

Our response: We appreciate the comment from the reviewer. We have added all the references in the article based upon the comment. Thanks.
